# Review of Technology-Supported Multimodal Solutions for People with Dementia

**DOI:** 10.3390/s21144806

**Published:** 2021-07-14

**Authors:** Majid Zamiri, Joao Sarraipa, Fernando Luis-Ferreira, Gary Mc Manus, Philip O’Brien, Luis M. Camarinha-Matos, Ricardo Jardim-Goncalves

**Affiliations:** 1NOVA School of Science and Technology and UNINOVA—CTS, 2829-516 Caparica, Portugal; jfss@uninova.pt (J.S.); flf@fct.unl.pt (F.L.-F.); cam@uninova.pt (L.M.C.-M.); rg@uninova.pt (R.J.-G.); 2Waterforf Institute Technology, Telecommunications Software and Systems Group (TSSG), X91 WR86 Carriganore, Ireland; gary.mcmanus@waltoninstitute.ie (G.M.M.); Philip.OBrien@WaltonInstitute.ie (P.O.)

**Keywords:** supportive technologies, supportive services, dementia, people with dementia (PwD), caregiver, PwD care

## Abstract

The number of people living with dementia in the world is rising at an unprecedented rate, and no country will be spared. Furthermore, neither decisive treatment nor effective medicines have yet become effective. One potential alternative to this emerging challenge is utilizing supportive technologies and services that not only assist people with dementia to do their daily activities safely and independently, but also reduce the overwhelming pressure on their caregivers. Thus, for this study, a systematic literature review is conducted in an attempt to gain an overview of the latest findings in this field of study and to address some commercially available supportive technologies and services that have potential application for people living with dementia. To this end, 30 potential supportive technologies and 15 active supportive services are identified from the literature and related websites. The technologies and services are classified into different classes and subclasses (according to their functionalities, capabilities, and features) aiming to facilitate their understanding and evaluation. The results of this work are aimed as a base for designing, integrating, developing, adapting, and customizing potential multimodal solutions for the specific needs of vulnerable people of our societies, such as those who suffer from different degrees of dementia.

## 1. Introduction

As life expectancy is noticeably increasing, and birth rate falling in many countries, there is a big shift in age distribution across the world towards an older population, particularly in developed countries [1,2]. Demographic aging of the world population indicates that more older adults are experiencing a variety of age-related problems, including dementia. Even though the causes of dementia have yet to be fully understood, it is reported that the mixture of family history, lifestyle, genes, and age can increase the risk of developing such disease. The fact is that dementia can occur for anyone, but the risk increases with age. As people get older, they become more prone to develop debilitating conditions, although dementia is not necessarily a natural part of aging [3].

Dementia is now one of the most prominent challenges that society is facing. As our world population rapidly ages, dementia is becoming one of the main social and health-related problems in the 21st century. The latest figures show that the chance of developing dementia at the age of 65 is high, and it can considerably increase for those over 85 [4]. Dementia has become a global problem, and it is reported that someone in the world develops dementia every 3 s. Furthermore, the prevalence and incidence of dementia in developing countries are higher [5].

Dementia is a profoundly life-changing condition, and its negative impacts are undeniable. For example, people with dementia (PwD) lose control of their lives, and the responsibility for their care shifts to caregivers and family members. Additionally, societies must provide the much-needed support services and facilities. As such, dementia needs proper and continuous healthcare as patients’ cognitive, physical, and functional abilities diminish over a period of years [6]. It is worth noting that dementia care is time-intensive, being a stressful and extremely emotional journey as well as a heavy burden. Furthermore, fewer family members have the time or the required experience to appropriately care for these older adults [7]. In addition, the current costs of dementia care are considerably high. For instance, in countries with the highest income, the costs of dementia diagnosis and treatment are more than those of heart disease and cancer, and these mostly stem from residential care. To face these problems and their challenging issues, there are some possible approaches, including resorting to the following:Informal caregivers—which include family, relatives, friends, or neighbors, who usually provide unpaid assistance and try to cope with the underlying physical or mental disabilities of PwD.Supportive technologies—which involve any item (e.g., piece of equipment, mobility device, or other specific technology) that can be used to increase, maintain, or improve the functional capabilities of PwD.Supportive care services (by formal caregivers)—involving government, medical partners, communities, social service organizations, authorities, online forums, dementia charities, or other institutional entities that can provide helpful information, guidance, and diverse care services for PwD and their informal caregivers.

Considering the cost and limitations of current caregiving practices, the application and adoption of supportive technologies represent a promising option. In this sense, assistive technologies combined with diverse care services become a key response to this emerging challenge of the 21st century. Therefore, in this study, the focus of attention is on supportive technologies and services, addressing the second and third above-mentioned approaches. 

Supportive technologies are basically designed to enable a patient, such as a person with dementia, to perform a set of tasks that she/he would otherwise be unable to do. Such technologies support PwD to live independently and also reduce the pressure on their caregivers. Supportive technologies and services can also enhance the easiness and safety with which the assistance can be performed. There is a lot of evidence that shows that PwD can live easier and better lives with dementia and take control of their condition if they could have access to appropriate technological support. Furthermore, the accessibility in real time to suitable technologies can benefit a person with dementia with regards to autonomy, wellbeing, and mental health in a much better way than depending solely on caregivers or family. Such technological support can, for instance, provide significant savings on long-term care costs, avoid dispensable admission to hospital or care home, reduce the potential safety risks (both inside and outside the home), facilitate communication between the PwD and caregivers, provide wellness and location monitoring opportunities, and more importantly improve the quality of life for PwD [1,8,9]. 

Complementarily, there are a wide variety of support services to assist PwD in different ways. Depending on which stage the affected person is on his/her life journey, different services would be more appropriate for them (e.g., social service, training, expert advice, care consultation, information provision) [1]. These supportive services can be provided by a range of organizations, e.g., health centers, health professionals, or individual caregivers. For instance, the Dementia Services Information and Development Centre (DSIDC) [10] is a center that offers three core professional services: (i) education and training, (ii) information and consultancy, and (iii) research services. Additionally, considering where the patient resides, he/she can benefit from local, national, or international supportive services. As an example of a supportive service at a national level, the Alzheimer’s Disease Research Centers (ADRCs) [11] at major medical institutions across the United States offer a range of services for families and patients affected by Alzheimer’s disease, including (a) help with diagnosis and medical management; (b) information provision about the disease, services, and resources; (c) opportunities for volunteers to participate in clinical trials and studies and patient registries; and (d) support groups and other special programs for volunteers and their families. 

Such supportive technologies and/or services are developed for utilization in nursing homes, patients’ homes, or even both. In our case (patient’s home and its surrounding environment), the smart home and ambient assisted living (AAL) each can share potential solutions. AAL provides an ecosystem in which different types of devices (e.g., computers, wireless networks, sensors, mobile phones, hardware, and software applications), services, processes, techniques, and ways help the person (mostly elderly and PwD) to live longer and independently for as long as possible. AAL programs and systems (e.g., technologies, and services) aid in making the lives of older adults safer, easier, comfortable, and, to some extent, self-dependent. As such, AAL initiatives can improve the performance of PwD in their daily life [2].

In a prior study [12] we conducted a survey to review a number of potential and available supportive technologies (with single functionality) that have applications for PwD. As an extension to that previous study and in consonance with [13], in this work we develop a literature review toward reaching three major objectives: (a) increasing our understanding of the latest findings in this realm, (b) identifying and documenting a set of supportive technologies (with multiple functionalities) and helpful services as a contribution to this field of study, and (c) using the results of this study as a foundation for developing an intelligent location monitoring system that can be customized to meet the specific demands of PwD (reducing the risks of wandering) and their informal caregivers (finding the location of PwD). 

For this study, we considered multimodal solutions that use a combination of two different, but complementary, means (supportive technologies and services) in order to enrich the outputs of this work. In addition, multimodal solutions, in our case, help in developing the scope of the study and provide wider opportunities for application. This approach is aligned with other works proposing approaches for collaborative multistakeholder care service provision [1]. It is also in line with the recommendations of a European strategic roadmap for information and communication technologies (ICT) and aging [9]. Thus, as an exploratory study, we attempt to identify and characterize a number of potential supportive technologies and services for PwD and their caregivers. These are then evaluated based on systematic reviews. 

The remainder of this article is structured as follows: In Section 2, the research method used for this study is briefly explained. In Section 3, the supportive technologies with multiple functionalities are explained. In Section 4, the identified supportive services are presented. In Section 5, the data collected for this study are analyzed. In Section 6, a discussion is developed about the findings of this survey. The paper ends with some concluding remarks and a brief look into possible future work in Section 7. 

## 2. Research Method

For this study, a systematic literature review was adopted in order to identify, analyze, and integrate the findings of multiple research works. To properly guide the study, the following research question is formulated: What kind of supportive technologies and services have potential application for improving the quality of life of PwD?

To answer this research question, the following hypothesis is adopted:
The quality of life of PwD improves if they can benefit from customized multimodal solutions.

With the intention of searching and choosing relevant (English language) articles for the survey, the main engineering and computer science databases (i.e., Scopus, Web of Science, IEEE Xplore, Compendex, and INSPEC) and psychology databases (i.e., PubPsych, PsycInfo, PTSDpubs, and PubMed) were searched. 

In database searching (for title, abstract, and body text), we used a variety of search terms and keywords (alone and in combination) such as “dementia”, “people with dementia”, “technologies”, “supportive technologies for PwD”, “assistive technologies for PwD”, “technology and dementia”, “ubiquitous healthcare technologies”, “pervasive healthcare technologies”, “smart homes”, “sensors”, “services”, “supportive services for PwD”, “services for dementia”, “supportive services for PwD and caregivers”, and “national and international care services for PwD”. Studies were included if they were original journal articles, surveys, book chapters, conference materials, and technical reports. We excluded studies that were not in the English language. Other document types, such as editorials, conference abstracts, position statements, and letters, were also excluded. 

Following initial searches, 255 publications within the period of 2010–2020 were identified for this study. After reviewing keywords, abstracts, and conclusions for relevance, 100 full papers were selected for reading. Narrowing the selection, a total number of 30 target papers (21 papers related to supportive technologies and 9 papers related to supportive services) were eventually included in the final analysis. Additionally, by using the above-mentioned keywords, 83 related commercial websites were also identified and reviewed to find further supportive technologies and services that were not addressed in the scientific papers. Having removed the websites containing unrelated solutions to our research topic, 15 relevant websites were finally selected (9 websites related to supportive technologies and 6 websites related to supportive services). Figure 1 demonstrates the selection procedures.

The selected papers and websites were then reviewed and analyzed according to the above-mentioned guiding research question. Afterward, the potential assistive technologies with multiple functionalities and supportive services and their characterizing information were extracted and summarized in two distinct tables (Tables 3 and 5, presented in Section 3 and Section 4, respectively). The assistive technologies with single functionality were first identified, analyzed, documented, and then published in our prior study as a complementary part of this overall research work [12]. 

In the following, an analysis of AAL systems is conducted to facilitate developing a classification system to categorize the main findings of this study (addressed in Tables 3 and 5). The literature shows that AAL systems have been classified under different taxonomies and accommodated in different groups and subgroups according to their primary functions. For example, as can be seen in Table 1, [14] classified AAL systems under three main groups, namely physical, cognitive, and social. The work described in [15] extended the number of classes to 11 main groups. In addition, [16] classified AAL systems under four groups and 27 subgroups.

After reviewing and considering the proposed classifications for AAL in the literature, and also inspired by previous related studies (e.g., the studies that are cited in Table 1), the authors focused on criteria to be taken into account for technology selection. The considered criteria for this purpose include alignment with AAL objects, alignment with the objectives of this study, product availability, functional capabilities, and technical features. Afterward, the authors proceeded to the next phase and proposed their own classifications (see Tables 3 and 5). These classifications stand on the application of open coding (aims at identification and categorization of the major concerning issues around the topic of this study to be then used for comparison). The names of the proposed groups and subgroups are taken and labeled while heeding the context and goals of this study and the defined research question, aiming to provide some directions for further developments. The proposed multimodal solutions can be potentially used as a guide in designing and developing an intelligent location and proximity monitoring system to improve wandering outcomes. Furthermore, the proposed solutions in this study can serve as a guiding principle to introduce and deliver customized, low-cost, and low-energy consumption technologies and supporting services with the intention to improve the safety and quality of life of PwD [12,13].

Given that, two complementary classifications are proposed in this study. The first one (see Table 3) is developed for the supportive technological solutions, and the second one (see Table 5) is for the supportive services. Each classification/table contains five main groups. The titles of the first four proposed groups in both classifications/tables are similar. That is, they are titled as cognitive, environmental, functional, and tele-information issues. However, the fifth groups in the proposed classifications for technological solutions and supportive services address the physiological and social issues, respectively. Furthermore, the titles of addressed subgroups for each group (in both classifications/tables) are proposed based on the most concerning issues in the identified papers. In Table 2, our proposed 6 groups are placed on top, and the similar groups and subgroups (from Table 1) that are in line with each respective group are listed below. For example, the cognitive issues (the first group in our classification) are also addressed in [14,16]. This conformity demonstrates that this study is in line with prior studies.

Taking into account the objectives of AAL, a brief description is provided for each proposed group in our classification:Cognitive issues—a category of mental health functions that affect cognitive abilities such as learning, thinking, perception, memory, and problem-solving.Environmental issues—a group of environmental factors that affect health and heath care conditions such as physical environment, indoor and outdoor conditions, safety, routines, climate changes, air pollution, and chemical pollution.Functional issues—functional status and capabilities that can be conceptualized as the ability to perform self-care, self-maintenance, and physical activities.Tele-information issues—these include collaborative telehealth and telemedicine care that provide long-distance health and health-related support for both PwD and caregivers, such as consultation, training, medical diagnosis, management, and care.Physiological issues—these deal with the mechanical, physical, and biochemical functions and the normal functions of living organisms and their parts such as brain, heart, and skin.Social issues—a group of social support actions and activities for both PwD and caregivers such as helping PwD with various daily tasks, social integration, and financial support.

## 3. Supportive Technologies with Multiple Functionalities

Supportive technologies in this context are a group of tools and healthcare devices that support and improve the quality of life of PwD. These technologies assist patients to be independent and even maintain active social engagements. Some technologies are also helpful for promoting the health and care of PwD, such as safety devices, memory aids and communication tools, technologies for smart homes, GPS trackers, and monitoring devices (for both body conditions and environment monitoring) [12,17]. 

In this study, a total of 30 potential supportive technological solutions, such as integrated systems, devices, sensors, wireless networks, and software applications, both basic and advanced, with multiple functionalities were identified. These technologies are from the literature (prior related studies) and complemented with relevant websites (to identify further technologies that were not cited in the scientific databases). They were then clustered into five main groups: (1) cognitive enhancement technologies, (2) environmental technologies, (3) functional technologies, (4) tele-information technologies, and (5) physiological technologies. These five groups were divided into some subgroups based on their own related features, characteristics, and functional capabilities that the identified technologies which support them may have. In the following, the proposed groups and respective subgroups are pointed out in a structured way where a letter represents a group (e.g., A) and a pair composed of letters and a number represents a subgroup (e.g., TA1—in this naming, T represents the classification for Technologies, A represents the first proposed group in this classification, and 1 represents the first subgroup of the first group in this classification). 

A.Cognitive enhancement technologies—used to develop human mental capacity.TA1. Capturing data from physical behaviors and emotional patterns;TA2. Support doing daily tasks;TA3. Tracking behaviors;TA4. Providing communication with others.B.Environmental technologies—used to measure single or multiple environmental contextual factors.TB1. Checking safety and generating alarm/alert;TB2. Controlling the conditions of the environment;TB3. Locating.C.Functional technologies—used to check and monitor the position, movements, activities, and performance of the PwD.TC1. Monitoring activities;TC2. Checking performance;TC3. Tracking hand function;TC4. Detecting fall;TC5. Checking motion and/or gait.D.Tele-information/healthcare technologies—used for the purpose of improving security and facilitate consulting. Such technologies can support virtual encounters between healthcare providers and patients.TD1. Telecare.E.Physiological sensing technologies—used to measure single or multiple physiological signs and basic metabolic parameters.TE1. Checking energy expenditure;TE2. Checking the quality of sleep;TE3. Checking body condition.

More detailed information is presented in Table 3.

Some of the identified and listed technological solutions (in Table 3) can be used for different groups of people (e.g., elders, children), and some are specifically for PwD. A brief explanation for each technology is provided below in order to facilitate the understanding of them. This explanation includes what the mentioned solutions are as well as their main functions and components. As such, relevant references are cited.

(1)*CareMedia* [18] is an automated video and sensor-based analysis system that allows the behavior of patients to be more accurately interpreted and monitored through intelligent browsing tools and filtered audiovisual evidence.
-Main functions: capturing audio and video data; monitoring environment and activities; and identifying, analyzing, and synthesizing the behavior of patients.-Main components: computer vision and machine learning technologies, miniature camera, galvanic skin response (GSR) sensors, and microphones.(2)*Cognitive Orthosis for Assisting Activities in the Home (COACH)* [19] is a prototype of an intelligent computerized system that helps PwD to perform activities of daily living with less dependence on caregivers.
-Main functions: monitoring and tracking the PwD during daily living activities.-Main components: artificial intelligence algorithms, digital video camera, charge-couple device, and hand-tracking bracelets.(3)*Mimamori-care system* [20] monitors the behavior of PwD in a group home facility. This system supports the following three issues: (a) quality of life of PwD, (b) quality improvement of mimamori-care, and (c) efficient improvement of care work.
-Main functions: monitoring the behavior of PwD, detecting their positions through several display screens, and providing communication with families and caregivers.-Main components: server, web browser, camera, display screens, integrated circuit (IC) tags (13.56 MHz passive RFID tag), and integrated circuit sensors (e.g., temperature sensor).(4)*Wearable and wireless camera system* [21] is a wearable technology for caregivers. It captures what happens between the caregiver and the PwD. This system helps family members and healthcare providers to understand the daily challenges that both PwD and caregiver face. Thus, it can enable the caregiver to learn more about effective ways of handling such challenges.
-Main function: capturing data related to what the caregiver sees, hears, says, and does. It also helps to monitor the activities and check the performance of PwD.-Main components: an outward-facing CMOS camera with a fish-eye lens, a MEMS microphone, battery, and garment.(5)*Physical activity monitor* [22] is a wearable technology in the form of an armband that offers some possibilities for the nutritional management of dementia sufferers.
-Main functions: evaluation of the relationship between daily energy expenditure and patterns of activity with energy intake of PwD. It includes predicting energy expenditure, sleep duration, and physical activities and checking body condition.-Main components: a triaxial accelerometer, a thermistor-based skin sensor, a proprietary heat flux sensor, a galvanic skin response sensor, and corresponding control software.(6)*Kognit* [23] is a kind of intelligent cognitive enhancement system that is concerned with artificial intelligence-based situation awareness, thus providing augmented cognition for the PwD. Kognit recognizes everyday objects, faces, and text that the PwD looks at.
-Main functions: supporting daily tasks, monitoring medication management, tracking behaviors, and checking body condition and performance of PwD.-Main components: pen gestures, video cameras, GPS, Bluetooth beacons, eye tracker, speech input, image analysis modules, and biosensors.(7)*Sensor-based in-home monitoring system, using remote web technologies* [24], is a multisensor system that is installed in a room and helps to monitor the PwD during the performance of daily activities. This system provides various cognitive aids to support the daily life of PwD.
-Main functions: detecting and monitoring the presence of PwD in the room, measuring physical activity, capturing image data, checking motions, and measuring duration of sleep.-Main components: ambient depth cameras; tags that attach to objects of interest, (e.g., watering can); plug sensors that attach to electronic devices (e.g., cooking appliances); sleep sensor placed underneath the mattress to measure sleep duration and interruptions; wearable wristwatch; and middleware that provides data retrieval, storage and analysis, and applications.(8)*Intelligent assistive technology (IAT)* [25] is a wearable system that detects and tracks significant moments based on patterns of physiological signal changes in PwD.
-Main functions: detecting salient physical and emotional events, recording the physiological signals from a device worn on the hand of PwD.-Main components: hardware (autonomic nervous system), triple point sensor, and custom software (event finder).(9)*Buddi* [26] is a wristband that is designed to be worn 24 h a day, so the user can always use it even in the shower or bath. Buddi helps the users to keep in touch with people they would like to connect with.
-Main functions: locating, detecting falls, and sensing the motions of user; creating alarms; and communicating with people who take care of the user.-Main components: alert buttons, accelerometer, fall sensor, and GPS.(10)*Ultra-wideband (UWB)* [27] is a system that attaches to a belt or wrist for monitoring the users (elderly persons) in their home.
-Main functions: monitoring of user’s behaviors, tracking gait, locating, detecting fall, and alerting.-Main components: reference anchor node (the main processing unit that controls other peripheral devices), system controller, tags (that are attached to the user and equipped with a micro-electromechanical system), accelerometer, environmental sensor, barometer, atmospheric pressure meter, and UWB transmitters.(11)*Smart carpet* [28] is an intelligent carpet for in-home monitoring of PwD.
-Main functions: tracking the behaviors of PwD nonintrusively, detecting motion and fall, locating (indoor), and alerting caregivers wirelessly.-Main components: mats or carpet tiles, each one having a pressure sensor.(12)*INdoor–OuTdoor Elderly CAring SystEm (NOTECASE)* [29] is a real-time tracking and monitoring system that the users (elderly, children, and PwD) can use in both indoor and outdoor environments.
-Main functions: monitoring the movements of the user, locating, sending alerts, and providing communication to people they like to contact.-Main components: windows phone (mobile device), wearable-enabled RFID tags to attach to the user, RFID reader that reads active RFID tags and communicates to network devices, Wi-Fi, and GPS.(13)*Wandering Detection Algorithm* [30] is a real-time system that automatically classifies wandering patterns (random, lapping, and pacing) or behaviors of PwD.
-Main functions: detection and classification of wandering patterns and behaviors, monitoring the movements of PwD, locating, and detecting spatial movements.-Main components: Global Positioning System (GPS), Wi-Fi, Bluetooth, integrated circuit/chip tag, radio frequency (RF) tag, radio frequency identification (RFID) tag for localization, RFID reader.(14)*ActionSLAM* [31] is a wearable indoor tracking system for home and workplace environments.
-Main functions: monitoring, locating, and checking movements of the users.-Main components: two inertial measurement units (IMUs) that attach to the foot and hip of the user, accelerator, laser line scans, and Wi-Fi.(15)*Indoor localization network* [32] is an indoor localization and activity monitoring system for caregivers to aid in the prioritization of surveillance to the PwD.
-Main functions: monitoring activities, locating, detecting the motions and falls, and tracking the behaviors of PwD.-Main components: static nodes that are used to determine a mobile node’s position (they have an on-board microcontroller processor and a radio transceiver module for wireless communication which will be placed at known locations within a building), mobile nodes that are incorporated within a wristwatch casing (to be carried out by PwD to locate their position and measure their motion activity), base node (showing the current position of mobile nodes), radio frequency tracking combined with motion and heading sensors.(16)*Nonintrusive pervasive computing model (using a wireless sensor network)* [33] is a kind of tracking system that can detect when PwD leave a safe location without supervision.
-Main functions: tracking the behaviors, locating, detecting falls, and generating alerts.-Main components: radio frequency identification (RFID) bracelet that attaches to PwD, passive infrared (PIR) sensors (measuring the infrared light emanating from objects and detecting the presence of heat from an object or body nearby), magnetometer sensors (to detect the change in direction of a magnetic field and recognize whether or not the doors are open), binary sensors (that reduce the processing overhead, allowing a faster system response), and motion detectors (that use PIR sensors).(17)*WearNET* [34] is a wearable technology including a distributed multisensor system able to collect a wide range of complex context information.
-Main functions: monitoring user’s activities, detecting motion, locating, and checking body condition and environment.-Main components: galvanic skin response (GSR) sensors, GSR electrodes, inertial navigation sensors (acceleration, gyroscope, and magnetic field sensors), environmental sensors, motion sensors, and GPS.(18)*SenSay* [35] is a context-aware system based on a mobile phone that improves the overall usability of the phone. It can modify its behavior based on user’s state and surroundings.
-Main functions: providing communication with people who the user would like to contact.-Main components: a mobile phone, a sensor box, voice and ambient microphone sensors, motion sensors, light sensors, and sensor module (collects physical sensor data).(19)*Wristband sensor* [36] is a wearable sensor-based system (wristband) that measures the stress level of PwD.
-Main functions: classifying events as “Stressed” and “Not stressed” for PwD (through detection of stress patterns) and then providing clinical information of behavioral patterns to the clinical staff and helping them in planning clinical interventions. These main functions are supported by collecting several parameters, typically using galvanic skin response (GSR), accelerometers (for capturing motion), skin temperature, and environment temperature and environment light measurement.-Main components: wristband sensor, skin sensor, and Discreet Tension Indicator (DTI-2) sensor.(20)*iTraq Nano* [37] is a location tracking device that works anywhere in the world.
-Main functions: reporting the accurate location of the user both outdoors and indoors.-Main components: accelerometer (detecting different types of motion), GPS, Wi-Fi, and Bluetooth.(21)*PocketFinder* [38] is a tracking device that can help users to know where they are and be alerted if they go too far.
-Main functions: assisting families and caregivers to find where the PwD (user) is.-Main components: location technologies (GPS, Cell-ID, Wi-Fi, Google Touch Triangulation) and Google Premier Mapping.(22)*Spy Tec Mini GPS Tracker* [39] is a tracking device that can track users and find their real-time location and if needed can send an alert by phone or email.
-Main functions: assisting caregivers and family members to find the accurate location of the user.-Main components: GPS and tracking software.(23)*SPOT GEN3* [40] is a tracking device for checking the location of a user.
-Main functions: tracking the behaviors of the user, locating, and communicating (through messages or email) with up to 10 contacts. It can also check all user’s motions.-Main components: SPOT messenger and GPS.(24)*GPS SmartSole* [41] is a wearable GPS tracker system that can fit easily into most adult shoes.
-Main functions: tracking the behaviors of users, locating, and sending alerts by text messages or email.-Main components: 2G cellular technology and GPS.(25)*Footprint* [42] is a GPS tracker device that is used within a telecare environment for elders to provide reassurance and safety.
-Main functions: tracking, locating, detecting falls, and alerting.-Main components: GPS or general packet radio service (GPRS), SOS button, and speaker.(26)*Wearable NFC wristband* [43] is a wristband rapid alert system that uses mobile sensing to locate PwD.
-Main functions: locating and sending notifications (when needed) to related service center personnel, nearby police station, and family members.-Main components: Near-field communication (NFC)-tagged wristband, encrypted chips, NFC sensors (that can be used in the cell phones of passersby to detect the NFC tag), GPS, and Hypertext Transfer Protocol Secure (HTTPS).(27)*Picture-Based Input Method Using Tapping on Wall Surfaces (PiTaSu)* [44] is a wearable and portable projection-based display system.
-Main functions: providing projected images on physical surfaces (e.g., wall), picture-based graphical user interface, and a simple tapping input interaction with the user interface. Thus, PwD can use interaction and communication (with tapping actions when necessary) by mobile device and ask for help anytime and regardless of place.-Main components: procams, Wi-Fi, accelerometer, Bluetooth, and wearable main unit/wearable PC.(28)*Mindme Watch* [45] is a digital smart alarm watch.
-Main functions: telling the user the time and date, how many steps he/she has taken, and the battery state. The watch can update its location around every 4 min. It can also detect falls and raise an alert.-Main components: SOS button, GPS, GPRS satellite technologies, tracking APP, and mobile APP.(29)*LESHP GPS Tr* [46] is a GPS tracker and emergency alarm system that can call or text three preprogrammed phone numbers with location details and other key information.
-Main functions: alerting location, tracking movements, setting digital boundaries for a specific area, and detecting the falls of users.-Main components: SOS button, fall monitor, GPS tracker, and two-way voice recorder.(30)*VTAM T-shirt* [47] is a wearable technology (T-shirt) that can communicate with a distance platform in emergency situations.
-Main functions: monitoring user’s activities, recording body temperature and respiratory rate, detecting falls, data transmission, and offering hands-free communication.-Main components: four smooth dry EKG electrodes (electrocardiogram), a shock/fall sensor, a breath rate sensor, two temperature sensors, GPS, Global System for Mobile Communications (GSM), motherboard, and belt.

This list is not exhaustive, as new technological solutions emerge continuously, but it can give a good overview of the current state of the art and trends in this area. Table 4 summarizes the main components used in each above-mentioned technology. 

## 4. Supportive Services

Supportive or care services for PwD have often been developed around specific needs (e.g., physical, mental, emotional, comfort, attachment, inclusion, and occupation needs). There are diverse support and care services to help PwD live more independently and comfortably throughout the various stages of their illness. It is interesting to note that some of those services are even provided for free. Existing services range from cognitive aids (e.g., timely diagnosis, assessment, and management) to information (e.g., counseling, education, and training), including home, community, and peer support; support for caregivers of PwD; and café-style support services.

In this study, 15 active supportive services are identified and selected, among others, from the literature (related prior studies) and relevant websites. For service selection, some criteria are considered, including alignment with AAL goals and practices, alignment with the objective of this study, having proactive technology-based approach, providing community-based support, supporting engagement, and delivering home care services. These services are classified into five main groups in the same way as the above classification for technological solutions. The classification for supportive services also used a similar rationale. However, the fifth group “physiological sensing technologies” used in the classification for technological solutions is replaced with “social services” in the classification for supportive services. The respective subgroups for each group, in this classification, are defined based on the specific services and assistance types that the identified services provide. In order to realize in which environments (indoor, outdoor, or both) the identified services can be delivered, the facet “type of care” is added to this classification. 

The group of services and respective subgroups in this classification are presented below (with the same structure that was used for supportive technologies, but in this case S represents the classification for services). 

A.Cognitive services—these focus on early diagnosis and also rehabilitation of cognitive, neurocognitive, and social deficits.SA1. Diagnosis and therapy;SA2. Cognitive support;SA3. Emotional support.B.Environmental services—these focus on improving the care environment and access to facilities and equipment.SB1. Finding the location of PwD;SB2. Access to centers;SB3. Providing meals and laundry.C.Functional services—these focus on physical activities and help PwD to maintain their mobility and functions.SC1. Physical activities.D.Tele-information/healthcare services—these focus on healthcare training and consultants.SD1. Telecare, training, and consultants.E.Social services—these focus on safeguarding and promoting welfare and social activities.SF1. Daily and nursing care;SF2. Social activities and events;SF3. Community support;SF4. Caregiver support;SF5. Financial support.

More detailed information is presented in Table 5.

An explanation for each addressed service is in Table 5, with the aim of enhancing the understanding of these services. It shall be noted that service provision often requires some supportive technology. As such, some of the described services might appear to have some overlap with the previous section. In fact, most of the technological solutions previously presented could originate or support new care services. Nevertheless, in the following list, the main aim is to illustrate types of services, irrespective of the underlying technology (if any).

(1)*Day care centers (DCCs)* [48] are center-based community care services, outside the patient’s own home, that mainly provide services such as daytime care, rehabilitation exercises, social activities and stimuli, as well as various events for PwD.(2)*Commonwealth Respite Carelink Centers (CRCCs)* [49] are regional or local support services that provide generic and specific information on dementia, counseling, and nursing care for PwD. CRCCs also provide caregivers with different services such as information, education, training, counseling, emotional support, and financial assistance.(3)*NHS and social care services* [50] are local council units that provide a package of care and fund services to meet the physical and/or mental health needs of PwD. These services include providing some equipment and home adaptations, help after coming home from the hospital, nursing in a care home, meals on the wheels, access to day centers, and laundry service.(4)*Cognitive stimulation therapy (CST)* [51] is a psychological intervention recommendation service for PwD that aims to provide an environment in which patients have fun, learning programs, and social relationships with other members of the community. This service helps PwD to strengthen their physical abilities and cognitive skills.(5)*Palliative care service* [52] is a specialized medical care service for PwD that focuses on providing relief from the symptoms and stress of dementia by means of early identification, assessment, and treatment of diagnosed problems such as physical, psychological, and spiritual.(6)*Farm-based day care (FDC*) [53] is a type of outdoor care service that uses farm resources (e.g., agricultural landscape and farming activities) to promote physical and mental health abilities for PwD. This service provides group collaboration between PwD, health and social workers, and farmers. FDC is linked to a healthcare institution and productive agricultural farms.(7)*Smart assistive living* [54] is a service that uses a wireless sensor network and broadband network connectivity to collect and integrate the environmental, physical, cognitive, and physiological data of PwD in order to provide telecare and telehealth services for them.(8)*Adult Day Health Care* [55] is a medical-supervised service for PwD with mental and/or physical impairment. Services include transportation, nursing, leisure activities, physical therapy, nutrition assessment, speech pathology, medical social services, occupational therapy, rehabilitation and socialization, psychosocial assessment, nursing evaluation and treatment, dental service, and coordination of referrals for outpatient health.(9)*Admiral Nurses* [56] is a professional nurse service that provides PwD and their caregivers practical support and help in a variety of ways. For example, they do therapeutic work, they are important sources of information about dementia; they bring about advanced assessment and prioritization of workload; and they can go through the diagnosis process, balance the needs of the caregivers and PwD, assess the impacts that dementia has on the PwD, deliver preventive care and health promotion, respond to changing behaviors associated with dementia, and promote best practices.(10)*MedicAlert Safely Home* [57] is a Canadian emergency response service that tries to identify PwD who are lost and assist in a safe return home (through a 24/7 emergency hotline). When called, the hotline specialists contact caregivers or family members to let them know the location and situation of PwD.(11)*Integrated community support for PwD* [58] is a support service for the complex and changing demands of PwD. The community provides a variety of support services including health and medical support, emotional and practical support, public education, caregiver support, care coordination, funding, and providing related assistive technologies.(12)*Cognitive Dementia and Memory Service (CDAMS)* [59] is a specialist service that provides medical and allied health consultations for both PwD and caregivers. Services include delivering clinical diagnosis, appropriate treatments, education, support, and information, as well as linking the PwD and caregivers to other services and community supports.(13)*Respite care* [60] delivers some services for both caregivers and PwD. Regarding caregivers, it provides caregivers a temporary rest from caregiving, while the PwD receive care in a safe environment. The services for PwD include help interacting with other PwD, providing a supportive environment, and participating in different activities that match the personal abilities and needs.(14)*National Dementia Support Program (NDSP)* [61] is an Australian Government initiative that directly supports PwD and their families and caregivers. The main services of NDSP include improving the understanding and awareness about dementia and empowering the families and caregivers to make decisions about the support services they access.(15)*Dementia Waikato* [62] is a regional and charitable service provider for PwD and their families and caregivers. Waikato delivers a variety of services including home visits and telephone support, community awareness and support, information, education, and training, as well as providing different social activities.

## 5. Analysis of Collected Data

In this section, the identified supportive technological solutions are briefly analyzed in order to assist in drawing helpful conclusions.

### 5.1. Supportive Technological Solutions

The studied technologies and systems are clustered into five main groups as addressed in Table 3. Given this classification and taking the results of this analysis into consideration it can be said that among the five proposed groups (cognitive enhancement technologies, environmental technologies, functional technologies, tele-information technologies, and physiological sensing technologies), the most addressed group (among all addressed groups in the considered studies and related websites) is the functional technologies group with 40% occurrence. It is followed by the group of environmental technologies with 27%, the group of cognitive enhancement technologies with 25%, and the group of physiological sensing technologies with 7%. The group of tele-information technologies (with 2%) is the least addressed one (in the considered references). The types and percentages of the defined groups for the identified technologies with multiple functionalities are illustrated in Figure 2. 

The addressed groups in Figure 2 are divided into some subgroups as listed in Table 3. To clarify the situation of these subgroups, their types and percentages are shown in Figure 3. 

Looking at these percentages, it can be said that: -The technological solutions that are designed to find the real-time location (locating) of PwD (TB3) with 18% and technologies for monitoring their activities (TC1) with 14% are the most addressed ones among the others (in the considered references).-Those technological solutions that are designed to control the conditions of the environment (TB2) and check the energy expenditure (TE1) of PwD, both with 1%, do not receive as much attention as the others (in the reviewed references).

### 5.2. Supportive Services

Taking into account the 15 supportive services addressed in Table 5, it can be clearly seen that the focus has been more on the issue of social services (in the majority of the considered references). Thus, this group has the highest percentage of cases with 49%, whereas the environmental services have the lowest percentage (8%) of cases. More detailed information about the types and percentages of these groups is shown in Figure 4.

To elucidate the situations, the types and percentages of the subgroups listed in Table 5 are given in Figure 5.

Given the adopted classification for supportive services and the percentages in Figure 4, it can be concluded that: -Services that deliver telecare, training, and consultation supports (SD1) have received the highest percentage of application (15%) among the considered references.-Services for supporting the caregivers of PwD (SF4) are the second most commonly addressed services with 12%.-The services that provide cognitive support (SA2), assistance in locating and finding PwD (SB1), and access to centers and laundry (SB2), all with 2% of occurrence, are the least represented services.

It should be added that some of the addressed services in Table 5 are provided for indoor care and some are provided for outdoor care. The analysis shows that among all 15 services listed in this table, the majority (13 services) are developed for outdoor care, 1 service is developed for indoor care, and 1 service is developed for both indoor and outdoor care. 

The rank orders that the addressed technologies and services have gained (based on our analysis in this study) might give an overview of what has been done and what needs further investigation, thus providing an indication of what the gaps are in this domain. For example, some literature [9,12] shows that supportive technologies still lack a significant base of multidisciplinary research. Furthermore, they lack the adoption of proper concepts and mechanisms for establishing collaborative communities or collaborative networks in support of PwD. On the other hand, this body of knowledge still suffers from big gaps in terms of understanding how dementia can be completely and efficiently supported by technologies and services. Additionally, there is a need for further work to identify what kind of roles such technologies and services can play and to what extent they can serve in the prevention and controlling of dementia [12,13,63].

## 6. Discussion

This section clarifies the added value of the survey done by introducing a new study, which addresses the main requirements and specifications for developing an intelligent location monitoring system for PwD. In this direction, this section, in relation to this new study, presents (a) the main considerations that are taken to achieve the objectives of such system development, (b) a comparison of the findings of the considered prior studies with the considered intelligent system to be developed, and (c) the potential application of findings of this new study. On that basis, in the following, Table 6 and Table 7 represent the key consideration for developing the proposed classifications (related to the identified supportive technologies and services) to be then used in developing the considered intelligent system. Given the above, the authors indicate eight types of considerations (with different importance values) for the defined subgroups for the technologies, namely:-It is not considered now (value = 0);-It is open for caregivers (value = 1);-It is open for PwD (value = 2);-It is open for both PwD and caregivers (value = 3);-It is considered for caregivers (value = 4);-It is considered for PwD (value = 5);-It is considered for both PwD and caregivers (value = 6);-It is highly important to be considered (value = 7).

As an example, in Table 6, the subgroup “TA1” is “considered for PwD” and is given the value 5. Taking into account the proposed technological subgroups listed in Table 3 and Table 6 gives more detailed information about the types of consideration and their assigned values for the considered intelligent system to be developed. 

To have a better view about the findings of and consideration in this new study, the details are presented by some illustrations in this section. Figure 6 allows a comparison between the proportions of the proposed “groups” for the identified supportive technologies with multiple functionalities in the considered prior studies and the considered intelligent system to be developed.

As can be seen in Figure 6, both the prior considered studies and the considered intelligent system nearly follow a similar pattern of consideration for the proposed technological groups. In addition, they both give high attention to the group of functional issues (TC). In considered intelligent system, the group of cognitive issues (TA) has received the most attention in comparison with the other four groups.

Figure 7 also illustrates the proportions of the proposed “subgroups” for the identified supportive technologies with multiple functionalities in the considered prior studies and the considered intelligent system.

As can be observed in Figure 7, the patterns of consideration for the proposed technological subgroups in both the considered prior studies and the considered intelligent system share some similarities. Apart from considered prior studies and among all the proposed subgroups, the considered intelligent system has given attention to: a.Tracking the PwD (TA3)—this helps them to navigate their ways. Therefore, some electronic tracking technologies such as GPS can be used in the considered intelligent monitoring system. Such technologies assist tracking, monitoring, and locating the PwD who are vulnerable to becoming lost.b.Safety of PwD (TB1)—this is highly important to be noted, particularly in the events of disorientation and wandering. It is very significant for family members and caregivers to ensure that the PwD return home safely. Thus, the considered intelligent system, by focusing on developing intelligent localization technologies, brings attention to the issues of monitoring journey routes of PwD, detecting occurrences of wandering, and alerting when these people deviate towards dangerous zones. Thus, it gives PwD a sense of independence and reduces the overall level of stress.c.Providing effective communication (TA4)—this includes some possibilities for ICT technologies to be used in the considered and developed intelligent system for monitoring various components of health-related factors such as physiological parameters and changes in activities. 

Taking into account the supportive services listed in Table 5, particularly the proposed subgroups of the services, Table 7 presents more detailed information about the types of consideration and their assigned values of importance for the considered intelligent system.

In the same manner as already described above in this section for the supportive technologies, in this part, a comparative analysis is presented for the supportive services. Figure 8 allows a comparison between the proportions of the proposed “groups” for the identified supportive services in the considered prior studies and for the considered intelligent system.

As depicted in Figure 8, both the prior studies and the considered intelligent system approximately follow a similar pattern of consideration for all the proposed groups of services except for the group of environmental issues (SB). SB emphasizes the issues of security and wandering of PwD. It should be noted that the group of social issues (SF) has equally received high attention in both the prior studies and the considered intelligent system.

Figure 9 shows the proportions of the proposed “subgroups” for the identified supportive services in the considered prior studies and the considered intelligent system.

As demonstrated in Figure 9, the patterns and considerations for almost all subgroups of supportive services in considered prior studies and the considered intelligent system are not close to each other. This is due to the considered intelligent system deliberately and purposefully focusing on two main issues: a.Finding the location of PwD (SB1)—this helps to locate them accurately. This issue plays an important role in dementia care and management.b.Community support (SF3)—this brings together a number of services (e.g., community-based services and social care services) and supportive entities (e.g., families, caregivers, relatives, friends, healthcare services, and voluntary organizations) around the agreed goal of supporting the protection of PwD from wandering.

As mentioned earlier, the emphasis of these services is given to promoting the quality of life and independence of PwD. Furthermore, these services aim at possibly and practically supporting PwD and their caregivers in the community toward creating an impact in this context.

The findings of this survey not only could be of broad use by researchers, developers, and designers in this realm but also are already used in the CARELINK project [12,13]. The core mission of the CARELINK project, in which this overall research was conducted, is to create smart and adaptive solutions for positively managing the wandering of PwD, thereby reducing stress for affected individuals and their caregivers [12,13,17]. The path of development follows applying potential supportive technologies for PwD and taking the advantages of complementary hardware (e.g., devices, sensors, embedded intelligence systems) and integrating them with appropriate supportive services, attempting to create an impact on the development of effective solutions. That is, by utilizing supportive technologies, such as smartphones and smart watches, and enriching them with tailored hardware (e.g., GPS, Wi-Fi, and Bluetooth), the idea aims to configure and customize solutions for various scenarios of real-life situations. The next step consists of carrying out trials of real-life instances to ensure that the proposed and developed solutions, as expected, can deliver a high-class system. Aimed solutions, however, will not provide a treatment or cure for PwD; rather, they come up with options targeting the improvement of quality of life and reducing the stress of both patients and informal caregivers.

As a conclusion, these findings were used as a base input for the CARELINK project. The main focused solutions were used in the process of monitoring the location and proximity of PwD. It should be added that the adopted method for conducting this survey and also some of the above-mentioned solutions (e.g., monitoring activities, community supports) should be used by authors in another project at hand, the FAITH project [64]. The FAITH project tries to develop a better model for mental health monitoring during disease and treatment for cancer patients to improve their quality of life and aftercare.

As a final point, it should be highlighted that all technological solutions and supportive services identified in this research have the potential to provide specific support for PwD, their caregivers, or both, in particular circumstances. The identification of more potential solutions with different characteristics and capabilities and the development of more related and specific classifications could guide scientists and developers working in this area.

As further work, we intend to conduct research into the potential complementary hardware (e.g., devices, sensors, chips, and microchips) and related embedded intelligence. 

## 7. Conclusions

Today, many countries across the globe are facing disproportionate growth of their aging population. It is a good thing that people are living longer, but it also becomes a problem when the care services are not sufficient for coping with this growth. There is also a substantial growth in the percentage of people developing dementia. Dementia is a set of syndromes that greatly influence cognition and other crucial functions of the brain. The economic, social, physical, mental, and emotional impacts of dementia on societies, healthcare systems, families, caregivers, and specifically affected individuals are of immense impact at different levels. Difficulties with day-to-day routines, communication, traveling, mobility, recognition, language, and memory can in turn lead to radical challenges. The more the problem progresses, the more special support and care the PwD needs. 

With the purpose of assisting those living with dementia in experiencing better quality of life, this research work reviews, identifies, classifies, analyzes, and documents a total of 30 potential supportive technological solutions with multiple functionalities and 15 supportive services. They are addressed in two tables and classified into some groups and subgroups according to the considered issues in AAL programs and objectives of this study. The results of this study can be used for both further investigation and development and as a foundation to provide and/or customize appropriate solutions for supporting PwD. It is expected that the proposed multimodal solutions that emerge from the findings of this study can:-Provide an effective contribution to this domain;-Help in responding to the pressing demand for identifying the ways that can assist in relieving the burden of dementia on sufferers and their caregivers;-Help in introducing supportive devices and services that increase the safety and independence of PwD;-Help in delivering a competitive intelligent location monitoring system;-Take a step in improving the quality of life of PwD.

This study is further expected to raise awareness about the existing supportive technologies and services that can empower users to better deal with not only the day-to-day realities of dementia but also other AAL situations.

## Figures and Tables

**Figure 1 sensors-21-04806-f001:**
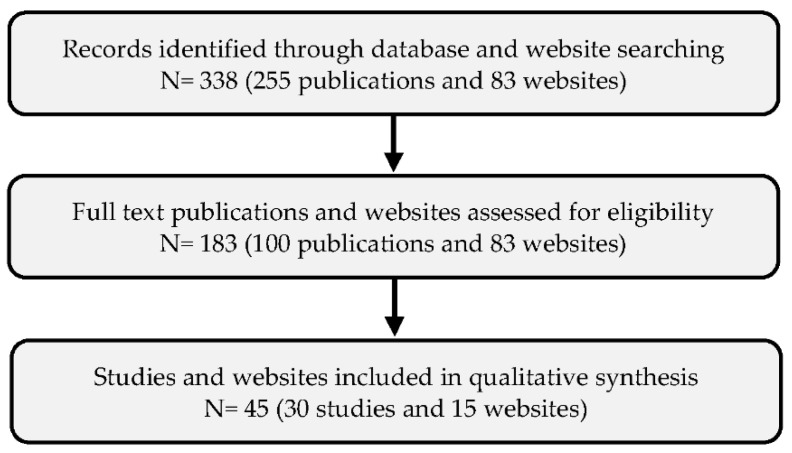
Procedures for selection of potential studies and websites.

**Figure 2 sensors-21-04806-f002:**
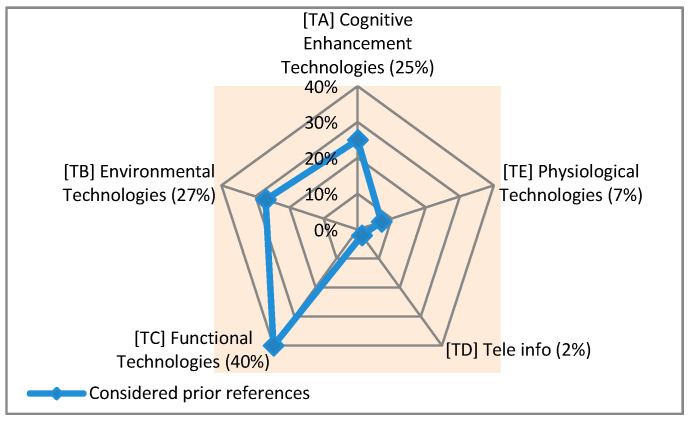
Types and percentages of proposed groups for the identified supportive technologies with multiple functionalities in the considered prior studies.

**Figure 3 sensors-21-04806-f003:**
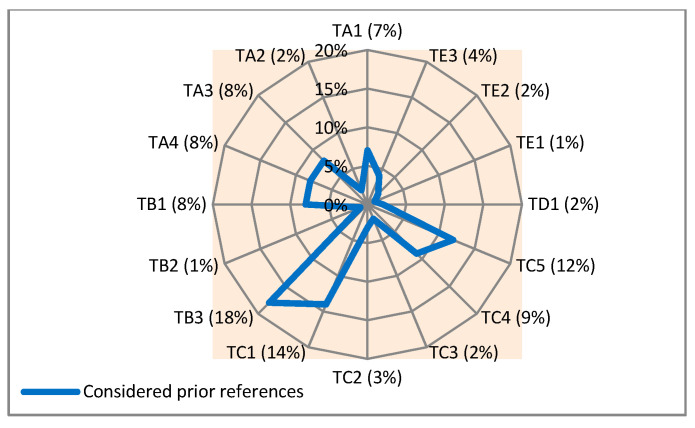
Types and percentages of proposed subgroups for the identified supportive technologies with multiple functionalities in the considered prior studies.

**Figure 4 sensors-21-04806-f004:**
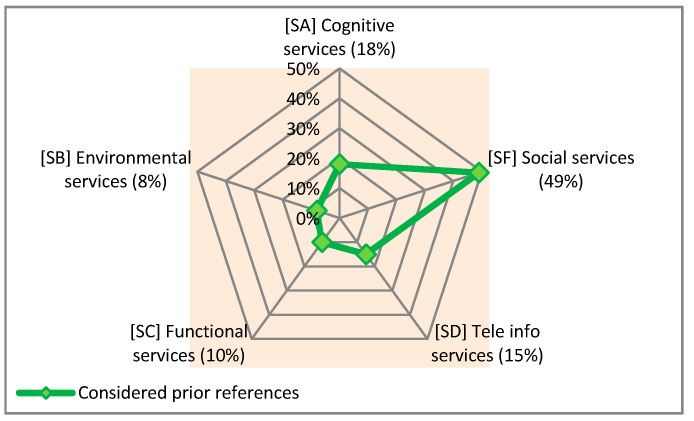
Types and percentages of proposed groups for the identified supportive services in the considered prior studies.

**Figure 5 sensors-21-04806-f005:**
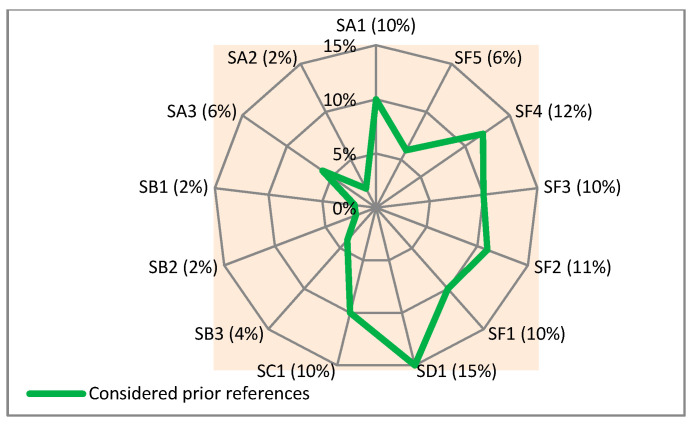
Types and percentages of proposed subgroups for the identified supportive services in the considered prior studies.

**Figure 6 sensors-21-04806-f006:**
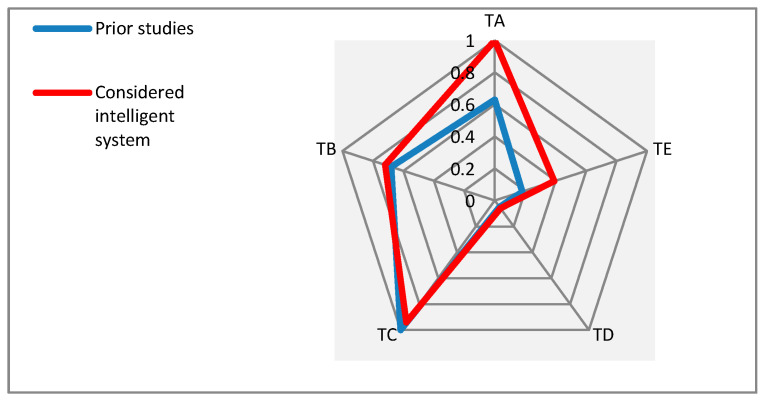
Comparing the proportions of proposed groups for the identified supportive technologies with multiple functionalities in the considered prior studies and the considered intelligent system.

**Figure 7 sensors-21-04806-f007:**
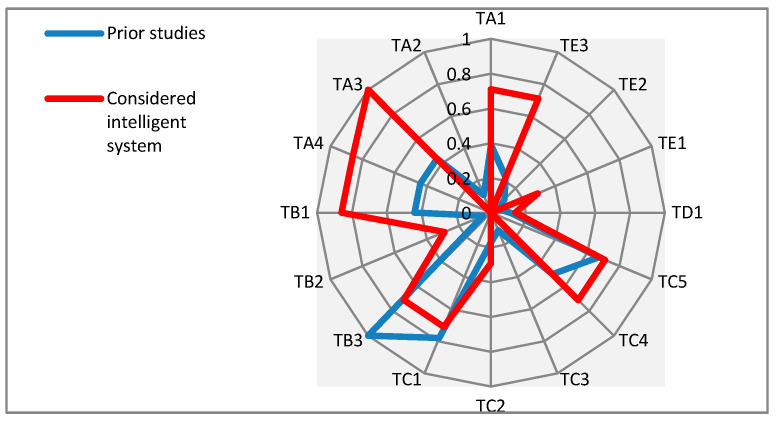
Comparing the proportions of proposed subgroups for the identified supportive technologies with multiple functionalities in the considered prior studies with the considered intelligent system.

**Figure 8 sensors-21-04806-f008:**
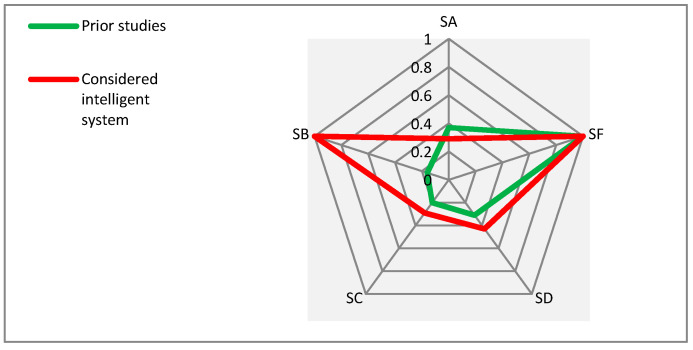
Comparing the proportions of proposed groups for the identified supportive services in the considered prior studies with the considered intelligent system.

**Figure 9 sensors-21-04806-f009:**
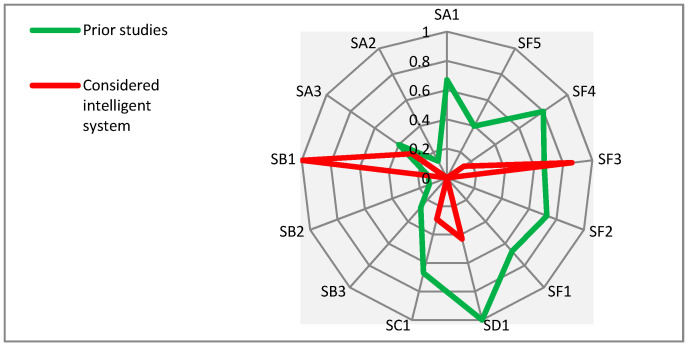
Comparing the proportions of proposed subgroups for the identified supportive services in the considered prior studies with the considered intelligent system.

**Table 1 sensors-21-04806-t001:** Proposed groups and subgroups for AAL technologies and services from three different studies.

No	Proposed Groups and Subgroups for AAL Systems from Three Different Studies											
[14]	Physical				Cognitive			Social				
[15]	Activity monitoring	Alerts	Communication	Emergency	Feedback support	Health monitoring	Navigation	Recreation	Social support	Standards	Specialized user interface	
[16]	Wearable	Smart Home		Smart Home						Robotic assistant		
		General health monitoring	Platforms	Wandering prevention tools (WPTs)		Electronic home control systems		Fall detection systems		Service and companion	Health	Intelligent physical movement aids
	----	- Specific health monitoring - Tele-performance - Activity of daily living (ADL) assistance	- Physical mobility assistance - Social inclusion and health	- RFID - Wi-Fi - Indoor WPT - Commercial WPT - Research-based WPT		- Intelligent home systems - Smart devices - Monitoring home automation sensors		- Orientation awareness system - Wearable fall detectors - Posture and movement awareness system		- Specific task robots - Cognitive orthotic robots - Daily activity reminder	----	----

**Table 2 sensors-21-04806-t002:** Conformity among the proposed classifications in this study with the addressed classifications in Table 1.

Conformity between the Classification Proposed in This Study and the Recommended Classifications in Table 1					
Proposed groups in this study					
Cognitive issues	Environmental issues	Functional issues	Tele-information issues	Physiological issues	Social issues
Similar proposed groups and subgroups addressed in Table 1					
- Cognitive [14] - Cognitive orthotic robots [16]	- Navigation [15] - Smart home [16] - Wandering prevention tools [16] - Electronic home control systems [16]	- Physical [14] - Activity monitoring [15] - Fall detection systems [16] - Intelligent physical movement aids [16] - Specific health monitoring [16] - Activity of daily living (ADL) assistance [16] - Physical mobility assistance [16] - Wearable fall detectors [16]	- Alert [15] - Feedback support [15] - Communication [15] - Tele-performance [16]	- Health monitoring [15] - Health [16]	- Social [14] - Social support [15] - Social inclusion and health [16]

**Table 3 sensors-21-04806-t003:** Identified supportive technologies with multiple functionalities and their groups and subgroups.

Identified Supportive Technologies and the Proposed Groups and Subgroups for Them																
Technological solutions/systems	A				B			C					D	E		
	Cognitive enhancement technologies				Environmental technologies			Functional technologies					Tele -Info technologies	Physiological sensing technologies		
	TA1	TA2	TA3	TA4	TB1	TB2	TB3	TC1	TC2	TC3	TC4	TC5	TD1	TE1	TE2	TE3
	Capturing data from physical behaviors and emotional patterns	Support doing daily tasks	Tracking behaviors	Providing communication	Checking safety and generating alarm/alert	Controlling conditions of environment	Locating	Monitoring activities	Checking performance	Tracking hand function	Detecting fall	Checking motion and/or gait	Telecare	Checking energy expenditure	Checking the quality of sleep	Checking body condition
1	**x**							**x**	**x**							
2			**x**					**x**		**x**						
3	**x**			**x**				**x**				**x**				
4	**x**							**x**	**x**							
5														**x**	**x**	**x**
6		**x**	**x**					**x**	**x**							**x**
7	**x**	**x**						**x**				**x**			**x**	
8	**x**									**x**		**x**				
9				**x**	**x**		**x**				**x**	**x**				
10					**x**		**x**	**x**			**x**	**x**				
11			**x**		**x**		**x**	**x**			**x**					
12				**x**	**x**		**x**	**x**								
13	**x**						**x**	**x**				**x**				
14							**x**	**x**				**x**				
15			**x**				**x**	**x**			**x**	**x**				
16			**x**		**x**		**x**				**x**					
17						**x**	**x**	**x**				**x**				**x**
18				**x**									**x**			
19	**x**												**x**			
20							**x**									
21							**x**									
22							**x**									
23			**x**	**x**			**x**					**x**				
24			**x**	**x**			**x**									
25			**x**		**x**		**x**				**x**					
26					**x**		**x**									
27				**x**												
28					**x**		**x**				**x**	**x**				
29							**x**				**x**	**x**				
30				**x**				**x**			**x**					**x**

**Table 4 sensors-21-04806-t004:** List of core components used in the identified supportive technologies with multiple functionalities.

No	Technologies	Core Components
1	CareMedia	Computer vision and machine learning technologies, miniature camera, GSR sensors, and microphones
2	COACH	Artificial intelligence algorithms, digital video camera, charge-couple device, and hand-tracking bracelets
3	Mimamori-care system	Server, web browser, camera, display screens, IC, tags, and integrated circuit sensors
4	Wearable and wireless camera system	CMOS camera, MEMS microphone, battery, and garment
5	Physical activity monitor	Triaxial accelerometer, thermistor-based skin sensor, proprietary heat flux sensor, galvanic skin response sensor, and corresponding control software
6	Kognit	Pen gestures, video cameras, GPS, Bluetooth beacons, eye tracker, speech input, image analysis modules, and biosensors
7	Sensor-based in-home monitoring system	Ambient depth cameras, tags, plug sensors, sleep sensor, wearable wristwatch, middleware, storage and analysis, and applications
8	Intelligent assistive technology	Hardware, triple point sensor, and custom software
9	Buddi	Alert buttons, accelerometer, fall sensor, and GPS
10	Ultra-wideband	Reference anchor node, system controller, tags, accelerometer, environmental sensor, barometer, atmospheric pressure meter, and UWB transmitters
11	Smart carpet	Mats or carpet tiles, each one having a pressure sensor
12	NOTECASE	Mobile phone, RFID tags, RFID reader, Wi-Fi, and GPS
13	Wandering Detection Algorithm	GPS, Wi-Fi, Bluetooth, integrated circuit/chip tag, RF tag, RFID tag, RFID reader
14	ActionSLAM	IMUs, accelerator, laser line scans, and Wi-Fi
15	Indoor localization network	Static nodes, mobile nodes, base node, radio frequency tracking combined with motion and heading sensors
16	Nonintrusive pervasive computing model	RFID bracelet, PIR sensors, magnetometer sensors, binary sensors, and motion detectors
17	WearNET	GSR sensors, GSR electrodes, inertial navigation sensors, environmental sensors, motion sensors, and GPS
18	SenSay	Mobile phone, sensor box, voice and ambient microphone sensors, motion sensors, light sensors, and sensor module
19	Wristband sensor	Wristband sensor, skin sensor, and DTI-2 sensor
20	iTraq Nano	Accelerometer, GPS, Wi-Fi, and Bluetooth
21	PocketFinder	GPS, Cell-ID, Wi-Fi, Google Touch Triangulation, and Google Premier Mapping
22	Spy Tec Mini GPS Tracker	GPS and tracking software
23	SPOT GEN3	SPOT messenger and GPS
24	GPS SmartSole	2G cellular technology and GPS
25	Footprint	GPS or GPRS, SOS button, and speaker
26	Wearable NFC wristband	NFC-tagged wristband, encrypted chips, NFC sensors, GPS, and HTTPS
27	PiTaSu	Procams, Wi-Fi, accelerometer, Bluetooth, wearable main unit/wearable PC
28	Mindme Watch	SOS button, GPS, GPRS satellite technologies, tracking APP, and mobile APP
29	LESHP GPS Tr	SOS button, fall monitor, GPS tracker, and 2-way voice recorder
30	VTAM T-shirt	EKG electrodes, shock/fall sensor, breath rate sensor, temperature sensors, GPS, GSM, motherboard, and belt

**Table 5 sensors-21-04806-t005:** Identified supportive services and their groups and subgroups.

Identified Supportive Services and Proposed Groups and Subgroups for Them														
Services	Type of care	A			B			C	D	F				
		Cognitive services			Environmental services			Functional services	Tele-info services	Social services				
		SA1	SA2	SA3	SB1	SB2	SB3	SC1	SD1	SF1	SF2	SF3	SF4	SF5
		Diagnosis and therapy	Cognitive support	Emotional support	Finding the location of PwD	Access to centers and equipment	Providing meals and laundry	Physical activities	Telecare, training, and consultants	Daily and nursing care	Social activities and events	Community support	Caregiver support	Financial support
1	Outdoor							**x**		**x**	**x**	**x**		
2	Outdoor			**x**					**x**	**x**			**x**	**x**
3	Indoor					**x**	**x**			**x**				**x**
4	Outdoor		**x**					**x**	**x**		**x**			
5	Outdoor	**x**												
6	Outdoor			**x**				**x**			**x**	**x**		
7	Outdoor								**x**					
8	Outdoor	**x**					**x**	**x**		**x**	**x**			
9	Outdoor	**x**							**x**	**x**			**x**	
10	Outdoor				**x**									
11	Outdoor	**x**		**x**					**x**			**x**	**x**	**x**
12	Outdoor	**x**							**x**			**x**	**x**	
13	Outdoor							**x**			**x**		**x**	
14	Outdoor								**x**					
15	In/Outdoor								**x**		**x**	**x**	**x**	

**Table 6 sensors-21-04806-t006:** Types of consideration and their given values for the considered intelligent system to be developed.

Cognitive Issues				Environmental Issues			Functional Issues					Tele-information issues	Physiological Issues		
TA1	TA2	TA3	TA4	TB1	TB2	TB3	TC1	TC2	TC3	TC4	TC5	TD1	TE1	TE2	TE3
Capturing data from physical behaviors	Support doing daily tasks	Tracking	Providing communication	Checking safety and generating alarm/alert	Controlling conditions of environment	Locating	Monitoring activities	Checking performance	Tracking hand function	Detecting fall	Checking motion and/or gait	Telecare	Checking energy expenditure	Checking sleeping	Checking body condition
It is considered for PwD (V = 5)	It is not now considered (V=0)	It is highly important to be considered (V = 7)	It is considered for both PwD and caregivers (V = 6)	It is considered for both PwD and caregivers (V = 6)	It is open for PwD (V = 2)	It is considered for PwD (V = 5)	It is considered for PwD (V = 5)	It is open for PwD (V = 2)	It is not now considered (V = 0)	It is considered for PwD (V = 5)	It is considered for PwD (V = 5)	It is open for caregivers (V = 1)	It is open for PwD (V = 2)	It is not now considered (V = 0)	It is considered for PwD (V = 5)

**Table 7 sensors-21-04806-t007:** Types of consideration and their given values for the considered intelligent system.

Cognitive Issues			Environmental Issues			Functional Issues	Tel- info Issues	Social Issues				
SA1	SA2	SA3	SB1	SB2	SB3	SC1	SD1	SF1	SF2	SF3	SF4	SF5
Diagnosis and therapy	Cognitive support	Emotional support	Finding the location of PwD	Access to centers and equipment	Providing meals and laundry	Physical activities	Telecare, training, and consultants	Daily and nursing care	Social activities and events	Community support	Caregiver support	Financial support
It is not now considered (V = 0)	It is not now considered (V = 0)	It is open for PwD (V = 2)	It is considered for PwD (V = 7)	It is not now considered (V = 0)	It is not now considered (V = 0)	It is open for PwD (V = 2)	It is open for both PwD and caregivers (V = 3)	It is not now considered (V = 0)	It is not now considered (V = 0)	It is considered for both PwD and caregivers (V = 6)	It is open for caregivers (V = 1)	It is not now considered (V = 0)

## Data Availability

Not applicable.

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
