# Peer review of "Review of Technology-Supported Multimodal Solutions for People with Dementia"

_sensors, 2021, doi:10.3390/s21144806_

Round 1

Reviewer 1 Report

The article is a good review of the existing technologies. I don't have any concerns or suggestions.

Author Response

Dear Reviewer 

Thank you very much for the effort and time that you took to review our paper.

Best Regards

Authors

Reviewer 2 Report

The paper surveys supportive technologies for people suffering from dementia. Based on scientific literature and web sites, the authors analyze 30 potential supportive technologies and 15 active support services.The study focuses on what the authors call "multimodal solutions", that they define as "two different, but complementary, means (supportive technologies and services). The work extends a previous study by the authors and is linked to the author's CARELINK project.

The paper is well written and provides for an interesting review of supportive technologies, and in a lesser degree, of services.

#1
The keywords used to search and select the reviewed literature for the survey are not clearly given. They should be provided in a clearer way, so that future studies can be compared with this one.  

#2
Which criteria were used to select the 15 supportive services? 
The supportive services analysis (section 5.2) are highly dependent on the selected supportive services. This might introduce bias in the analysis.

#3
The paper often refers the CARELINK project, and even dedicates one section (Section 6) to the project. This seems odd in a review/survey paper.

#4
The paper would benefit to have a table to highlight the use of core technologies (e.g., GPS, Wi-Fi,...) by each project analyzed within the context of "supportive technologies". This is done within the text that describe each project (line 279 and beyond), but a table would make this information much more accessible.

Observations
============
1) Avoid tables split over two pages, as it makes for a difficult read.

2) Table 3: green rectangles might be difficult to distinguish in a black-and-white printout. Moreover the table is split over two pages, making for a difficult read.

Typos
=====
line 152: "for **his** survey" -> *this*
line 195: listed in **bellow** -> below
line 228: software **application** -> **applications**

line 227-231: the sentence is too long. Revise.

line 405:  (to detect the change of direction of a magnetic field -> missing
")"

Author Response

Dear Reviewer 

Thank you very much for the effort and time that you took to review our paper.

We also thank you for providing very helpful comments. We tried to cover all your comments. Please find the annex.

Best Regards

Authors

Reviewer 3 Report

Thepaper is well written and structured, but an additional check is required in order to fix some remaining spelling and grammar issues. However, its main weakness is that it remains only partly aligned with the journal's domain, it presents the state of the art without an in-depth analysis, making the paper read more like a project deliverable instead of a finalized research effort. Some additional issues, in more detail:

  • Inclusion criteria for solutions and services are not made clear. Authors state that 100 works (30 systems and 15 services) were selected based on an extensive search of online repositories for research results but the size of the original data set remains unclear.
  • Tables should probably be reorganized so that text is horizontally oriented.
  • Tables should also not break across pages. Figure captions should appear on the same page as the figure itself.
  • I don't find it clear how the authors reached the taxonomy presented in Tables 1&2. Was it similar to open coding?
  • I believe limiting the search to the realm of research literature presents a threat to validity, as commercial systems might exist that remain unexplored by existing studies. This is especially true for the results reported as part of Section 5.
  • Section 6 reads more like a deliverable for the CARELINK project than an independent research paper.

Author Response

(The authors gave the same response as above.)

Round 2

Reviewer 3 Report

The revised version of the paper is significantly improved.